# Oligometastatic Head and Neck Cancer: Challenges and Perspectives

**DOI:** 10.3390/cancers14163894

**Published:** 2022-08-11

**Authors:** Houda Bahig, Shao Hui Huang, Brian O’Sullivan

**Affiliations:** 1Department of Radiation Oncology, University of Montreal, Montreal, QC H2X 3E4, Canada; 2Department of Radiation Oncology, Princess Margaret Cancer Centre, University of Toronto, Toronto, ON M5G 2M9, Canada; 3Department of Otolaryngology-Head and Neck Surgery, Princess Margaret Cancer Centre, University of Toronto, Toronto, ON M5G 2M9, Canada

**Keywords:** distant metastasis, oligometastasis, head and neck cancer, treatment, prognosis

## Abstract

**Simple Summary:**

Oligometastasis represents a disease state and an opportunity for cure when metastases emerge. Emerging evidence supports that most head and neck cancer patients with oligometastatic disease are likely to benefit from curative intent local ablative therapy if appropriate selection criteria are applied. Biomarkers to predict development of oligometastasis, as well as to identify which patients could benefit from a radical intent approach, are under investigation. This review summarizes recent knowledge about the characteristics, investigational efforts, and evidence for local ablation regarding oligometastasis in head and neck cancer. We also describe the challenges and opportunities in patient selection and discuss the role of radiotherapy and immunotherapy combinations to enhance anti-tumor immunity.

**Abstract:**

A minority of patients with metastatic head and neck squamous cell carcinoma (HNSCC) present with oligometastatic disease. Oligometastasis not only reflects a disease state, but might also present an opportunity for cure in the metastatic setting. Radical ablation of all oligometastatic sites may confer prolonged survival and possibly achieve cure in some patients. However, substantial debate remains about whether patients with oligometastatic disease could benefit from curative intent therapy or whether aggressive treatments expose some patients to futile toxicity. Optimal selection of patients, carefully balancing the currently known prognostic factors against the risks of toxicity is critical. Emerging evidence suggests that patients with a limited burden of disease, viral-related pharyngeal cancer, metachronous metastasis and lung-only metastasis may benefit most from this approach. Efforts are underway to identify biomarkers that can detect oligometastasis and better select patients who would derive the maximum benefit from an aggressive radical approach. The combination of radiotherapy and immunotherapy promises to enhance the anti-tumoral immune response and help overcome resistance. However, optimization of management algorithms, including patient selection, radiation dose and sequencing, will be critical in upcoming clinical trials. This review summarizes recent knowledge about the characteristics and investigational efforts regarding oligometastasis in HNSCC.

## 1. Introduction

Distant metastasis was once considered uncommon in head and neck squamous cell carcinoma (HNSCC) due to the overwhelming risk of loco-regional failure in previous decades. However, the development of more effective management of the primary tumor and neck has now uncovered a relatively strong but variable risk prediction of distant metastasis as a function of anatomic site of origin, stage, and association with certain biological characteristics, especially viral disease etiology. Distant metastasis can occur at initial presentation as de novo metastatic disease (M1) or metachronous metastasis after initial treatment. Generally, there are two types of distant metastasis based on the extent of metastatic lesions and pace of growth: polymetastasis and oligometastasis. The latter isdefined as a state of a limited number of metastatic lesions confined to a single or limited number of organs [1]. Oligometastasis more frequently manifests in viral-related pharyngeal cancer (e.g., Epstein–Barr virus (EBV)-positive nasopharyngeal carcinoma and human papillomavirus (HPV)-positive oropharyngeal carcinoma) compared to non-viral related HNSCC [2]. In contrast to non-viral related HNSCC, where distant metastasis often follows or may coincide with locoregional failure, HPV-positive and EBV-positive distant metastasis often occurs in isolation without locoregional failure, suggesting that pre-clinical micro-metastasis may already exist at the time of diagnosis. It is important to point out that oligometastasis versus polymetastasis often represent a kinetic manifestation of distant metastatic disease. If left without treatment, many oligometastases can progress and become polymetastases. Conversely, with treatment, some polymetastases can become oligometastases. This is especially important in the era of immunotherapy.

Differentiating oligometastasis from polymetastasis is useful because different therapeutic strategies are needed. In recent years, there has been increasing interest in the concept of metastasis-directed local ablation in the context of oligometastatic and oligoprogressive disease across various cancer histologies, from different primaries, and at different secondary cancer sites. This is supported by recent prospective randomized phase II trials providing empirical evidence for improved progression free survival and overall survival with radical local treatment to all tumor sites, including HNSCC [3,4,5]. For patients with polymetastatic disease, immunotherapy alone or in combination with chemotherapy has become standard-of-care first line therapy [6,7]. Despite the recent advancements in metastatic HNSCC, the majority of patients present primary resistance to immunotherapy, while patients with initial benefit eventually develop secondary resistance [8,9]. When patients relapse after first line treatment, subsequent options are limited and include non-platinum chemotherapy, single agent immune checkpoint inhibitor monotherapy, platinum-based chemotherapy in platinum-naïve patients, participation in a clinical trial or best supportive care, with outcomes being generally dismal [10]. For patients with oligometastatic disease, increasing evidence shows that treatment intensification involving aggressive local ablation of metastatic sites may achieve prolonged survival, and sometimes cure [11,12]. Studies have reported long-term survival or even cure in viral-related oropharyngeal carcinoma [2,13] and nasopharyngeal carcinoma [14,15,16] following aggressive ablative treatment using surgery or radiotherapy with or without systemic agents.

This review summarizes recent knowledge about the characteristics and investigational efforts regarding oligometastasis in HNSCC. More specifically, we examine the evidence for local ablation of oligometastatic lesions derived from HNSCC. We also describe the challenges and opportunities in patient selection and discuss the role of radiotherapy and immunotherapy combinations to enhance anti-tumor immunity.

## 2. Defining Oligometastatic HNSCC

Most tumors progress in an orderly manner: from local disease, to regional nodal involvement, followed by hematogenous dissemination of tumor cells to distant organs/sites, i.e., distant metastasis. Oligometastatic disease represents an intermediate disease state between local and widespread metastasis (Figure 1). It often has a relatively indolent nature compared to widespread dissemination. However, without effective treatment, it can eventually become widespread. Although the term is somewhat pragmatic, it is also evolving, and is currently defined as metastatic cancer of limited disease burden, usually characterized by ≤5 clinically detectable lesions [11,17]. The concept of aggressive local ablation of oligometastasis (by surgical resection or radiotherapy) in order to achieve prolonged survival, and sometimes cure, was first proposed by Hellman and Weichselbaum in 1995 [11,12]. Oligometastatic disease can be synchronous, meaning that it is identified at the time of initial cancer diagnosis, or metachronous, implying that it develops after a certain interval (6 months is the typical definition in the literature, although UICC/AJCC uses a “4-month” window for metachronous situations) following initial locoregional treatment [18,19]. The latter is sometimes referred to as oligorecurrence (Figure 1). In contrast, the term oligoprogressive disease is loosely defined as progression to a limited number of sites after a systemic therapy that has resulted in relative disease stability, including a partial or complete response [20]. It has been hypothesized that oligoprogression results from tumor heterogeneity, whereby progression is observed in drug resistant subclones in a small number of sites [21,22]. 

## 3. Aggressive Local Ablation in Oligometastatic HNSCC

The use of metastasis-directed ablation has gained significant momentum in the aftermath of recent phase II trials, demonstrating that a subset of patients with oligometastatic disease appear to have improved progression free survival and overall survival following local ablation of all metastatic sites. Mounting evidence has stimulated official guidelines from international oncological associations such as the European Society for Therapeutic Radiology and Oncology (ESTRO) and the American Society for Radiation Oncology (ASTRO) to recognize local ablative therapy as an option for patients with oligometastatic disease [23]. In HNSCC specifically, the current evidence supporting the role of local ablation of oligometastatic disease is limited to a small number of retrospective studies [24,25,26,27,28]. The two primary local ablation modalities for curative-intent of metastasis include surgery and radiotherapy, specifically stereotactic ablative radiotherapy (SABR). SABR is defined as a highly conformal image-guided radiotherapy technique allowing for precise delivery of a high ablative dose of radiotherapy in a small number of fractions (typically 1 to 10) [29]. The use of SABR as a radical approach in selected cases of oligometastatic disease is particularly advantageous given its non-invasive nature, its excellent local control (of metastases) rates above 80% in most series [30,31], and its safety with <5–10% risk of grade ≥3 toxicities [32,33,34,35]. In addition, some pre-clinical and clinical evidence suggests that SABR may play a synergistic role in combination with immunotherapy by stimulating the immune response, which will be discussed in Section 6 [36].

In 2015, a meta-analysis of 11 studies including 387 patients with HNSCC with a controlled primary who underwent surgical resection of metachronous pulmonary metastasis showed a 5-year overall survival of 29% [26]. While two-thirds of patients had resection of a single nodule, one-third had multiple nodules, up to a maximum of six. Poor prognostic factors in this cohort included the presence of cervical lymph node metastasis at initial diagnosis, oral cavity primary site, incomplete pulmonary resection, and the presence of multiple pulmonary nodules. Several small subsequent retrospective studies (sample size ranged from 27 to 82 patients) of oligometastatic patients from HNSCC (58–100% involved the lungs) treated with surgery or SABR have reported overall survival rates as high as 75% at 1-year and 40–50% at 5-years [24,25,27,28,37]. Median overall survival was around 2 years in most series, but reached 47 months in some reports with exclusive lung metastasis [28]. More recently, the randomized phase II SABR-COMET trial (NCT01446744) included 99 patients with 1–5 metastases (of whom 10% had HNSCC) who were randomized to a standard of care versus SABR to all sites of disease, and showed a significantly improved 5-year overall survival with local ablation (42% vs. 17%, *p* = 0.006) [3]. Another recent phase II trial evaluating the role of SABR in patients with 1–5 metastases from various cancer histologies included 147 patients, of whom 10% had HNSCC [38]. For the entire cohort, median overall survival was 42 months and 5-year overall survival was 43%, while for the HNSCC subgroup, median survival was 18 months and 5-year overall survival was 42% [38]. However, the highly selected criteria for patient inclusion in these studies highlight the need for more robust prospective data focused on patients with HNSCC.

## 4. Selecting Optimal HNSCC Patients for Aggressive Ablation

Currently, the decision to radically treat a patient with oligometastatic HNSCC relies on careful selection based on age, co-morbidity, disease-free interval, known prognostic factors, feasibility and safety of local treatment, as well as institutional expertise (Figure 2). It is important to emphasize case-by-case selection for appropriateness and careful multidisciplinary discussions at the tumor board. Although severe toxicity remains rare, combining several irradiation sites with a systemic treatment will increase the risk of adverse sequelae. Pasalic et al. reported a 17% pulmonary toxicity rate, such as pneumonitis, but no grade 3 or higher toxicities [26]. Informed patient consent should include careful counselling about potential risks versus benefits of the approach. Favorable prognostic factors, as well as the feasibility and safety of local ablation of all sites of disease, must be considered to minimize unnecessary toxicity from local ablation. It remains unclear which patients benefit best from a radical approach compared to upfront palliative intent systemic therapy. However, based on retrospective data, patients with a smaller number of metastatic lesions, lung metastases, controlled locoregional disease or presence of virally associated (HPV or EBV) HNSCC have improved prognosis and therefore may be most suitable for an aggressive ablative approach [2,21,39,40].

### 4.1. Disease Burden

The number of metastatic lesions, together with the disease burden, is linked to cancer outcomes, including oligometastatic cancer in general, as well as in HNSCC specifically. In the context of surgical resection of HNSCC pulmonary metastases, the presence of multiple pulmonary nodules significantly decreases the survival probability for oligometastatic disease [40,41]. Fleming et al. reported decreased efficacy of radical metastasis-directed treatment associated with systemic therapy with an increasing number of metastases in metastatic HPV-positive HNSCC; for 1, 2–4 and ≥5 metastasis, median OS was 41.2, 17.2 and 10.8 months, respectively (*p* = 0.007) [42]. In a proposed treatment algorithm for patients with metastatic HNSCC, Tang et al. proposed that patients with ≤3 metastatic lesions benefit best from curative intent treatment, whereas patients with >3 metastatic lesions may benefit best from upfront systemic treatment [43]. In addition to the number of lesions, tumor volume is also an important consideration. In a cohort of patients with oligometastatic disease from any primary tumor site treated with SABR, Rusthoven et al. showed a significant difference in 5-year overall survival for differently sized SABR-treated metastases, with a 5-year local control rate of 100% for smaller lesions compared to 77% for those >3 cm [44]. However, the adverse survival outcomes with higher disease burden may not necessarily translate into a lack of benefit from local ablation since disease progression may still be modified after local treatment in these patients. The on-going phase III COMET-10 trial (NCT03721341), which randomizes patients with 4 to 10 oligometastasis from any solid tumors to SABR to all oligometastatic sites versus standard approaches, will assess the overall survival benefit from ablative therapy in this population with higher disease burden. Another interesting yet provocative area of research is the use of SABR beyond oligometastatic disease to achieve a delay in cancer progression with the goal of improvement in overall survival. This approach is currently being investigated in the Ablative Radiation Therapy to Restrain Everything Safely Treatable (ARREST) trial (NCT03880565) [45]. Hopefully, the results from the trial will better inform the interplay between the benefit of local ablation in relation to disease burden in metastatic cancer in the future.

### 4.2. Virus-Related Pharyngeal Cancer

Nasopharyngeal carcinoma holds a unique place in the evolution of these treatment approaches. It is more than 20 years since investigators in Europe, North America and Asia independently reported long term survivors following aggressive multimodality approaches involving chemotherapy with or without radiotherapy, and/or surgery [14,16,46,47]. Therefore, EBV-related nasopharyngeal carcinoma represents the first head and neck cancer site where a demonstrable cure appeared possible in some patients with distant metastasis. A growing body of evidence supports an aggressive treatment strategy for the primary tumor in the presence of synchronous oligometastatic disease with the possibility of prolonged survival. In a recent phase III randomized trial, locoregional radiotherapy combined with chemotherapy in patients with synchronous metastatic nasopharynx carcinoma improved overall survival in chemotherapy-sensitive patients (2-year overall survival 76.4% vs. 54.5%, *p* = 0.004) [48]. Shen et al. reported the results of a retrospective study of 312 patients with nasopharynx carcinoma metastatic to the bone exclusively, and found that the number of metastatic lesions (≤3 vs. >3 lesions), spine involvement (an adverse effect), and primary tumor-treatment approach (chemoradiation vs. chemotherapy or radiation only) were independent prognostic factors of overall survival [49]. Patients treated with chemoradiation had a 5-year overall survival of 57% compared with 11% in those who received palliative treatment. In another retrospective study of 263 patients diagnosed with metastatic nasopharynx carcinoma, patients with single-organ metastases or ≤5 metastatic lesions had a 5-year overall survival of 39% compared to 7% in those with multiple-organ metastases or >5 lesions [50]. In the latter study, treatment of the primary tumor with radical intent radiotherapy was also found to be a favorable prognostic factor for overall survival [50].

The possibility for cure in the metastatic setting has now extended to HPV-related oropharyngeal carcinoma in recent years. It is well established that HPV-positive oropharyngeal carcinoma is associated with improved overall survival and disease-free survival compared to their HPV-negative counterparts [51]. Although the actuarial rates of distant metastasis may not be significantly different [52,53], HPV-positive distant metastasis often manifests later [54,55], and can exhibit different characteristics including involvement of multiple organs and unusual sites [2] (e.g., brain, intra-abdominal and pericardial lymph nodes, duodenum, pancreas, spleen and kidney). In contrast to HPV-negative counterparts, HPV-positive distant metastasis often occurs without locoregional failure, suggesting occult metastasis may have occurred at the time of diagnosis and initial treatment. In addition, late onset distant metastasis has been reported in several retrospective studies [2,54,55,56]. For example, Huang et al. [56] reported two p16-positive oropharyngeal carcinoma patients with detectable p16-positive lung metastatic lesions more than 5 years after initial treatment; Sinha et al. reported a HPV-positive oropharyngeal carcinoma patient who developed lung metastases 8.8 years after treatment [55]. Besides the pace of distant metastasis manifestation, the characteristics of HPV-positive distant metastasis are also different. Huang et al. classified distant metastasis into two distinct phenotypes: a disseminating phenotype with “explosive” character with numerous metastatic lesions occupying almost entire organ(s) that developed over a relatively brief time period, and a relative “indolent” phenotype. The latter often manifested as “oligometastasis”, and is amenable to local ablative treatment such as surgery or modest to high dose radiation [56]. Long-term survival after distant metastasis in HPV-positive oropharyngeal carcinoma patients has been reported by several authors [2,13,55,56,57]. Based on a cohort from the Princess Margaret Cancer Centre, Huang et al. reported that five out of six HPV+ patients with lung oligo-metastasis were still alive with stable disease beyond 2-years after salvage procedures for distant metastasis (chemotherapy: three; surgical resection: two; radiotherapy: one) [56]. The different pattern of recurrence is also highlighted by another study by Sinha et al., reporting that among 66 patients with metastatic oropharynx carcinoma, locoregional disease was present in 52% of patients with HPV-negative disease, compared with 25% in the HPV-positive disease (*p* = 0.02) [55]. While a definitive metastasis-directed approach was attempted in 12% of HPV-positive and 27% of HPV-negative patients, all HPV-negative disease progressed or resulted in death within 2 years. In contrast, progression free survival after distant metastasis diagnosis was as high as 20% at 2-years in HPV-positive patients [55]. Similarly, Lee et al., showed that patients with metachronous oligometastatic HPV-positive oropharyngeal carcinoma benefit from initial metastases-directed therapy compared to upfront palliative intent systemic therapy, with a median overall survival not reached with definitive treatment vs. 40.7 months with systemic therapy [38]. In another series, among eight patients with metastatic HPV-positive oropharyngeal carcinoma, three underwent ablation of lung metastasis and remained free of disease 4–5 years after, raising the real possibility of cure in some patients [58].

## 5. Combined Immunotherapy and Radiation

The combination of local ablation with emerging systemic therapies in order to stimulate the immune response and enhance systemic response is an active area of research. At the cellular level, local radiation may trigger immunogenic cell death, which can promote systemic inflammation and immune-mediated activation of antigen-presenting dendritic cells and cytotoxic T cells, and ultimately anti-cancer immunity [59,60,61,62,63]. Pre-clinical [59,60,64] and early clinical [61,62,63] data suggests that the local use of radiotherapy in combination with immunotherapy can induce antigen release and T-cell activation, which can enhance the local and systemic effects of immunotherapy. However, radiation is a potential “double-edged sword” in the immunotherapy paradigm. It has been reported that, while radiation enhances antitumor immunity, it also induces an immunosuppressive response [65]. Most trials that have assessed the combination of radiotherapy and immunotherapy have involved single site irradiation and evaluated for an abscopal response in the untreated lesions distant from the irradiated site. In this setting, a pooled analysis from two randomized phase II trials in metastatic non-small cell lung cancer showed that pembrolizumab plus radiotherapy is associated with an improved (abscopal) response rate compared to pembrolizumab alone (42% vs. 20%) [66]. Other phase II trials have failed to demonstrate an abscopal effect from the combination of immunotherapy and radiotherapy to a single or small number of metastatic sites in polymetastatic disease [67,68]. Of relevance, a recent trial by the Memorial Sloan Kettering group randomized patients with polymetastatic HNSCC to single site SABR, but failed to show an abscopal effect from the combination of nivolumab and SABR, with an overall objective response rate of 34.5% with nivolumab alone vs. 29.0% with nivolumab and SABR, *p* = 0.86 [68]. In contrast to single site irradiation, several on-going studies are currently evaluating the role of multi-site irradiation (NCT03827577, NCT04944914, NCT03391869, NCT04402788). The hypothesis behind multisite irradiation in combination with immunotherapy involves enhancing treatment synergy through reduction of disease burden, increased radiation-induced immune response from a diverse repertoire of infiltrating T cells while maintaining local control in lesions likely to result in significant morbidity if left untreated [69]. Better understanding and optimization of radiation parameters, in a particular dose, fractionation, and sequence is critical in the design of future clinical trials. In addition, oligoprogression while on immunotherapy may result from resistant tumor clones, differences in tumor microenvironments or an immune adaptation resulting in acquired resistance [21]. Current management of patients presenting resistance to immunotherapy in the form of oligoprogression remains controversial and is underpinned by lack of evidence to guide decision making. In the specific context of tumors that are oligo-refractory to immunotherapy, the rationale for the use of SABR relies on controlling those lesions that progress on systemic therapy, while keeping immune pressure with the same systemic therapy strategy on the residual responding lesions. There are several on-going prospective randomized trials looking at the role of SABR for oligoprogression in lung cancer (NCT04405401; NCT03256981; NCT04485026; NCT03256981), renal cancer (NCT04299646), breast and lung cancers (NCT03808662), prostate cancer (NCT04141709) and multiple histologies (NCT02756793). Among these, the on-going Canadian Stereotactic Radiotherapy for Oligo-Progressive Metastatic Cancer (the STOP Trial) (NCT02756793) is a randomized phase II comparison (54 patients) assessing the progression free survival of SABR in oligoprogressive disease from any primary site (including head and neck cancer). In addition, the role of high and low dose radiation in oligoprogressive HNSCC is also being tested in a single arm prospective trial from the Dana-Farber Cancer Institute (NCT03085719). The Stereotactic Ablative Radiotherapy for Oligo-Progressive Disease Refractory to Systemic Therapy in Head and Neck Cancer (Suppress-HNC) Trial (NCT04989725) currently assesses the role of SABR in patients with HNSCC oligoprogressive to 1–5 sites while on immunotherapy. Table 1 summarizes the current studies evaluating combined radiotherapy and immunotherapy in oligometastatic and oligoprogressive HNSCC. These trials are needed to better clarify the benefits of treatment and improve patient selection.

## 6. Optimizing Surveillance Protocols

As noted earlier, oligometastatic HNSCC is detected either at diagnosis (synchronous distant metastasis) or at follow-up after initial locoregional treatment (metachronous metastasis). Lung is the most frequent metastatic site, accounting for two-thirds of distant metastases, followed by bone (22%) and liver (10%) [2,51]. While the exact prevalence is unknown, synchronous oligometastatic HNSCC is likely rare. Staging of all newly diagnosed locally advanced HNSCC involves pre-treatment chest CT and/or fluoro-deoxy glucose positron emission tomography (FDG-PET) scanning. In patients with more advanced locoregional disease (T3–4 and N1 or more), the use of FDG PET is usually favored as it was reported to upstage M0 to M1 disease in 9% and alter treatment in 14% of newly diagnosed previously untreated HNSCC in a previous randomized controlled trial [70].

It is estimated that one-third of metachronous distant metastases following initial curative treatment are oligometastases [71,72], and mostly occur within 3 years of initial cancer diagnosis; however, as previously discussed, delayed distant metastasis beyond 5 years has been reported in a small proportion of HPV-positive oropharyngeal carcinoma cases [2]. Patients with oligometastasis often do not show obvious signs/symptoms. Guidelines for imaging surveillance for metachronous metastasis are not well established and are poorly adapted for the detection of oligometastatic disease. In fact, with the exception of patients benefiting from follow-up chest CT for micronodules uncovered at initial diagnosis or routine screening low dose chest CT if they have a substantial history of smoking [73], the cost-effectiveness of routine imaging surveillance to detect distant metastasis is not defined. Whether routine chest CT can detect more distant metastasis in the oligometastatic state that may benefit from early local ablative intervention should be explored. This is especially the case among patients at risk of distant metastasis, such as hypopharyngeal primary origin, large tumors, extensive lymphadenopathy, extranodal extension, contralateral or low-level lymph nodes, or poor histological differentiation [74,75,76]. Finally, patients receiving systemic treatment who manifest oligoprogressive disease are typically routinely monitored by CT or PET scans every 6 to 12 weeks for assessment of disease response and timely detection of progression, and occasionally oligoprogression.

The current lack of surveillance imaging protocols in the post-locoregional treatment setting may translate to an underestimation of the prevalence of metachronous oligometastasis and missed opportunities for aggressive ablative treatment. Furthermore, the main caveat of the current standard of care imaging modalities is their inability to distinguish actual oligometastasis from microscopic polymetastatic disease. In fact, only macroscopic lesions detected radiologically are considered in the definition of oligometastasis. Whether widespread pre-clinical micro-metastases already exist at the time of oligometastatic detection by these imaging approaches, and whether patients with microscopic polymetastasis still benefit from local ablation of macroscopic oligometastasis, remains unclear. Predictive biomarkers derived from quantitative image analysis (radiomics) [77,78] and liquid-based genomics (cell free DNA or circulating tumor cells) [79,80] are potential surveillance methods of high interest given their non-invasive nature. Radiomic signatures, which consist of extraction and analysis of quantitative features from radiologic images, have been correlated with cancer control outcomes in HNSCC [78,81,82]. Liquid biopsy refers to the analysis of body fluids (blood, saliva, etc.) to detect and characterize cancer cells [83]. Among the most widely investigated, circulating tumor cells (CTC) refers to cells that have shed into the vasculature or lymphatics from a primary tumor [84,85], while circulating free DNA (cfDNA) in HNSCC includes circulating tumor DNA (ctDNA; released by tumor cells) and viral DNA (HPV or EBV DNA) [86,87]. Early results suggest that post-treatment plasma and salivary HPV ctDNA can predict disease recurrence in oropharynx carcinoma and may constitute a useful surveillance method [88,89]. In a recent study on 93 patients with HPV-positive oropharynx carcinoma, the presence of combined salivary and plasma HPV DNA after radical treatment was 91% specific and 70% sensitive in predicting recurrence within 3 years [80]. A multicenter study of 1076 HPV-positive oropharyngeal carcinoma from 108 institutions in the United States showed that ct-HPV DNA was able to identify 96% cases with occult recurrence [90]. Therefore, emerging evidence supports the promising role of these biomarkers for early cancer detection, tumor mutational burden evaluation and post-treatment surveillance, and are currently being investigated [80,91]. In fact, several trials (e.g., NCT03942380 and NCT02245100) are currently on-going, exploring role of cfDNA in disease surveillance for HNSCC.

## 7. Conclusions

As supported by growing evidence, oligometastasis not only represents a disease state, but also an opportunity for cure. In HNSCC, emerging evidence supports that most patients with oligometastatic disease are likely to benefit from curative intent local ablative therapy if appropriate selection criteria are applied. Patients with low disease burden and viral-related pharyngeal cancer appear to be the most likely candidates to achieve long-term disease control after local ablation. However, the benefit of local ablation may also lie in slowing disease progression in other patients at various levels within the spectrum of oligometastatic disease. Efforts are underway to identify biomarkers to predict development of oligometastasis, as well as to identify which patients could benefit from a radical intent approach. Until further prospective and randomized supportive evidence is available, balancing favorable clinical prognostic factors and potential benefit from aggressive local treatment against the risk of futile toxicity remains a reasonable approach in suitable patients. On-going clinical trials will further inform about the role of radiotherapy to enhance local and systemic anti-tumoral immune responses in combination with immunotherapy.

## Figures and Tables

**Figure 1 cancers-14-03894-f001:**
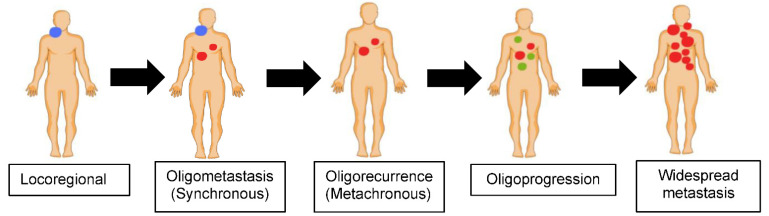
Metastatic disease spectrum.

**Figure 2 cancers-14-03894-f002:**
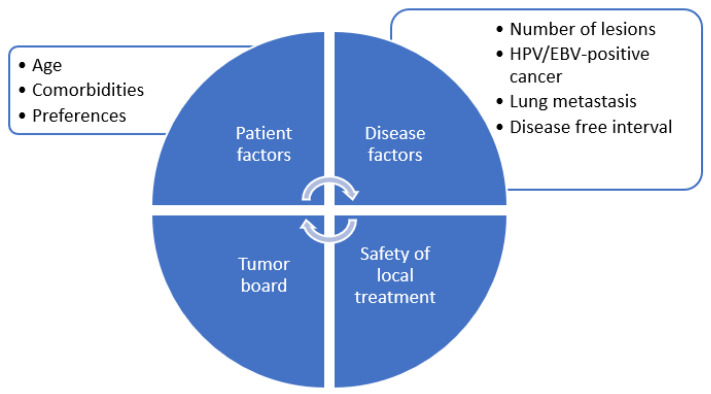
Approach to patient selection for radical treatment in metastatic head and neck cancer.

**Table 1 cancers-14-03894-t001:** Ongoing studies of combined SABR and systemic treatment in metastatic HNSCC.

Name	ID NCT	N	Group	Design	Intervention	Disease Type	Primary Outcome	Status
IMPORTANCE	NCT03386357	130	Erlangen-Nürnberg	Phase II randomized	Pembrolizumab +/− SABR 36 Gy/12 fx to 1–3 metastasis	Metastatic HNC	Best response	Recruiting
N/A	NCT04862455	60	MDACC	Phase II single arm	NBTXR3, RT + Pembrolizumab	Recurrent or metastatic HNC	PFS	Recruiting
OMET	NCT03070366	78	GORTEC-2014-04	Phase II randomized	Chemotherapy +/− SABR	Oligometastatic HNC (1–3 mets)	OS without QoL deterioration	Recruiting
SABR-COMET-3	NCT03862911	297	BCCA	Phase III randomized	SOC +/− SABR	Any oligometatstaic cancer (1–3 mets)	OS	Recruiting
SABR-COMET-10	NCT03721341	159	LHSC	Phase III randomized	SOC +/− SABR	Any oligometatstaic cancer (4–10 mets)	OS	Recruiting
Suppress-HNC	NCT04989725	46	CHUM	Phase II randomized	IO +/− SABR	Oligoprogressive	PFS	Recruiting
OZM-088	NCT03283605	35	CHUM	Phase I-II single arm	Durvalumab/Tremelimumab	Oligometastatic HNC (2–10 mets)	Toxicity (phase I) PFS (phase II)	Closed
LM-HNSCC	NCT05136768	50	Chinese Academy	Phase II single arm	Sintilimab/chemotherapy/SABR	Oligometastatic HNC (1–10 mets)	PFS	Recruiting
oligoRARE	NCT04498767	200	EORTC 1945	Phase III randomized	Continue current systemic therapy	Rare oligometastatic cancer	OS	Recruiting

Abbreviation: SABR: stereotactic ablative body radiotherapy, HNSCC: head and neck squamous cell carcinoma, HNC: head and neck cancer, OS: overall survival, PFS: progression-free survival, QoL: quality of life, DOR: duration of disease response, TPIL: time to progression of initials lesions, TPNL: time to progression of new lesions, TPNRL: time to progression of non-irradiated lesions, DCR: disease control rate, ORR: objective response rate, LC: local control, mos: months, DSS: disease-specific survival.

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
