# Peer review of "Oligometastatic Head and Neck Cancer: Challenges and Perspectives"

_cancers, 2022, doi:10.3390/cancers14163894_

Round 1
Reviewer 1 Report
Dear authors,
This manuscript conducted a comprehensive review of oligometastatic head and neck squamous cell carcinoma (HNSCC), in relation to definition, prognostic effects of metastasis-directed ablation, patients selection criteria, combination of local ablation with systemic therapy and patient surveillance protocols.
Here are some comments and possible suggestions.
Section 1:
This review first introduced that distant metastasis in HNSCC was recognized with the development of more effective management, and made it clear what is synchronous metastasis, metachronous metastasis, polymetastasis and oligometastasis. Authors then presented the different metastatic characteristics in viral-related and non-viral-related HNSCC, and introduced the role of viral infection in metastatic patterns. Then authors emphasized the importance of differentiating oligometastasis from polymetastasis and presented the therapeutic strategy differences between the two. Finally, authors introduced that the treatment intensification involving aggressive local ablation of metastatic sites may achieve prolonged survival, and sometimes cure for patients with oligometastasis.
Section 2:
The authors gave a clear definition of oligometastatic disease (synchronous and metachronous [oligorecurrence]) and oligoprogressive disease, and presented the hypothesized origin of the latter. Then the authors explained that metastasis-directed local ablation is gaining interest.
Possible suggestions:
Line 86: Like that for oligoprogressive disease, it might be better to provide a hypothesized origin of oligometastatic disease.
Lines 92-97: This part might be more suitable for introduction part rather than in the definition of oligometastatic HNSCC.
Section 3
The authors made it clear that metastasis-directed ablation has shown promising efficiency for patients with oligometastatic disease first, and then presented the fact that although guidelines are recognizing local ablative therapy as an option for patients with oligometastatic disease, for HNSCC these evidences are limited. Then they introduced the two primary local ablation modalities for curative-intent of metastasis, especially stereotactic ablative radiotherapy (SABR) in detail (definition and advantages). Next, the authors provided previous evidences from meta-analysis, retrospective studies and randomized trials showing the benefits patients with oligometastatic disease HNSCC may gain from curative-intent treatment. They also introduced the challenge of selection criteria for patient inclusion for receiving these treatments, which introduced the next section.
Section4:
The authors first summarized the current selection criteria, reason for selection (toxicity) and the challenges. Then they dissected it in detail from different aspects, focusing on disease burden (number of metastatic lesions and tumor volume) and virus infection status.
Virus infection part began with the long-term survival of EBV-related nasopharyngeal carcinoma (NPC) patients with distant metastasis. Then the authors discussed HPV-related oropharyngeal cancer (OPC), presenting different metastasis manifestations between HPV-positive and HPV-negative OPC. Next, the authors presented the classification of HPV-positive OPC distant metastasis, treatment and corresponding survival of these patients. Finally, the authors provided the radical intent radiotherapy for synchronous oligometastatic NPC.
Possible suggestions:
Lines 239-255: It might be better located after Line 195, with necessary modifying the sentences. These two parts are both related to NPC.
Section 5
The authors first summarized that oligometastatic HNSCC could be synchronous and metachronous. Then they introduced the necessity of pretreatment imaging for staging newly diagnosed locally advanced HNSCC and its implication for possible M stage upstaging and treatment decision changes in more advanced ones. Besides, 1/3 of metachronous distant metastasis were oligometastasis and provided their onset timing. Next, the authors stated that the guidelines for imaging surveillance for metachronous metastasis have not been established and surveillance imaging protocols are limited by providing the implications of the imaging technique clinically, and summarized the caveats of standard of care imaging modalities. Finally, the authors came up with the concept that biomarkers derived from radiomics and liquid-based genomics are potential surveillance methods and provided relevant evidences.
Possible suggestions:
Line 259: The most common metastatic site seems better located in introduction. Besides, this section might be better located at the end of the review or just after the definitions of oligometastatic disease (section 2). Other sections (section 3, 4, 6) are all treatment-related and it might be better if those sections are connected tightly.
Section 6
The authors first stated that combination of local ablation with systemic therapies are of promising future. Then they provided the results of ongoing trials exploring combination of radiotherapy with immunotherapy, both form the mechanism aspect and prognosis aspect (singe site irradiation and multisite irradiation). Then authors presented the challenges: radiation parameters need to be optimized, and the management of patients presenting resistance to immunotherapy in the form of oligoprogression remains controversial.
Author Response
Reviewer 1:
This manuscript conducted a comprehensive review of oligometastatic head and neck squamous cell carcinoma (HNSCC), in relation to definition, prognostic effects of metastasis-directed ablation, patients selection criteria, combination of local ablation with systemic therapy and patient surveillance protocols.
Here are some comments and possible suggestions.
Section 1:
This review first introduced that distant metastasis in HNSCC was recognized with the development of more effective management, and made it clear what is synchronous metastasis, metachronous metastasis, polymetastasis and oligometastasis. Authors then presented the different metastatic characteristics in viral-related and non-viral-related HNSCC, and introduced the role of viral infection in metastatic patterns. Then authors emphasized the importance of differentiating oligometastasis from polymetastasis and presented the therapeutic strategy differences between the two. Finally, authors introduced that the treatment intensification involving aggressive local ablation of metastatic sites may achieve prolonged survival, and sometimes cure for patients with oligometastasis.
Section 2:
The authors gave a clear definition of oligometastatic disease (synchronous and metachronous [oligorecurrence]) and oligoprogressive disease, and presented the hypothesized origin of the latter. Then the authors explained that metastasis-directed local ablation is gaining interest.
Possible suggestions:
Line 86: Like that for oligoprogressive disease, it might be better to provide a hypothesized origin of oligometastatic disease.
Response: Thank you for the suggestion. We have added the following sentences to address the reviewer’s comment:
“Most tumors progress in an orderly manner: from local disease, to regional nodal involvement, followed by hematogenous dissemination of tumor cells to distant organs / sites, i.e. distant metastasis. Oligometastatic disease represents an intermediate disease state between local and widespread metastasis (Figure 1). It often has a relatively indolent nature compared to widespread dissemination. However, without effective treatment, it can eventually become widespread metastasis.”
Lines 92-97: This part might be more suitable for introduction part rather than in the definition of oligometastatic HNSCC.
Response: We agree with the reviewer. We have now moved the following sentence into the Introduction (Line 65-70).
Section 3
The authors made it clear that metastasis-directed ablation has shown promising efficiency for patients with oligometastatic disease first, and then presented the fact that although guidelines are recognizing local ablative therapy as an option for patients with oligometastatic disease, for HNSCC these evidences are limited. Then they introduced the two primary local ablation modalities for curative-intent of metastasis, especially stereotactic ablative radiotherapy (SABR) in detail (definition and advantages). Next, the authors provided previous evidences from meta-analysis, retrospective studies and randomized trials showing the benefits patients with oligometastatic disease HNSCC may gain from curative-intent treatment. They also introduced the challenge of selection criteria for patient inclusion for receiving these treatments, which introduced the next section.
Section4:
The authors first summarized the current selection criteria, reason for selection (toxicity) and the challenges. Then they dissected it in detail from different aspects, focusing on disease burden (number of metastatic lesions and tumor volume) and virus infection status.
Virus infection part began with the long-term survival of EBV-related nasopharyngeal carcinoma (NPC) patients with distant metastasis. Then the authors discussed HPV-related oropharyngeal cancer (OPC), presenting different metastasis manifestations between HPV-positive and HPV-negative OPC. Next, the authors presented the classification of HPV-positive OPC distant metastasis, treatment and corresponding survival of these patients. Finally, the authors provided the radical intent radiotherapy for synchronous oligometastatic NPC.
Possible suggestions:
Lines 239-255: It might be better located after Line 195, with necessary modifying the sentences. These two parts are both related to NPC.
Response: We agree with the reviewer and have relocated this paragraph earlier, specifically after “EBV-related nasopharyngeal carcinoma represents the first head and neck cancer site where demonstrable cure appeared possible in some patients with distant metastasis.”
Section 5
The authors first summarized that oligometastatic HNSCC could be synchronous and metachronous. Then they introduced the necessity of pretreatment imaging for staging newly diagnosed locally advanced HNSCC and its implication for possible M category upstaging and treatment decision changes in more advanced ones. Besides, 1/3 of metachronous distant metastasis were oligometastasis and provided their onset timing. Next, the authors stated that the guidelines for imaging surveillance for metachronous metastasis have not been established and surveillance imaging protocols are limited by providing the implications of the imaging technique clinically, and summarized the caveats of standard of care imaging modalities. Finally, the authors came up with the concept that biomarkers derived from radiomics and liquid-based genomics are potential surveillance methods and provided relevant evidences.
Possible suggestions:
Line 259: The most common metastatic site seems better located in introduction. Besides, this section might be better located at the end of the review or just after the definitions of oligometastatic disease (section 2). Other sections (section 3, 4, 6) are all treatment-related and it might be better if those sections are connected tightly.
Response: We agree with the reviewer. To make the paper flow better, we have moved this section to the end of review just before Conclusion.
Section 6
The authors first stated that combination of local ablation with systemic therapies are of promising future. Then they provided the results of ongoing trials exploring combination of radiotherapy with immunotherapy, both form the mechanism aspect and prognosis aspect (singe site irradiation and multisite irradiation). Then authors presented the challenges: radiation parameters need to be optimized, and the management of patients presenting resistance to immunotherapy in the form of oligoprogression remains controversial.

Reviewer 2 Report
This is an interesting review which summarizes recent data regarding the risk of oligometastases in head and neck cancer patients with squamous cell carcinoma. A new approach to this problem based to an accurate review of current literature and analysis of survival results in patients with limited metastatic disease justifies the publication in Cancers in present form.
Author Response
Response: Thank you for your positive comment.
